# Dielectric Properties Investigation of Metal–Insulator–Metal (MIM) Capacitors

**DOI:** 10.3390/molecules27123951

**Published:** 2022-06-20

**Authors:** Li Xiong, Jin Hu, Zhao Yang, Xianglin Li, Hang Zhang, Guanhua Zhang

**Affiliations:** 1State Key Laboratory of Advanced Design and Manufacturing for Vehicle Body, College of Mechanical and Vehicle Engineering, Hunan University, Changsha 410082, China; lixiong@hnu.edu.cn (L.X.); jinhu8@hnu.edu.cn (J.H.); 2Guangdong Fenghua Advanced Technology Holding Co., Ltd., Zhaoqing 526060, China; 3State Key Laboratory of Advanced Material and Electronic Components, Zhaoqing 526060, China; 4School of Physics and Chemistry, Hunan First Normal University, Changsha 410205, China; lixl@idtu.cn; 5Key Laboratory of Applied Surface and Colloid Chemistry, Ministry of Education, School of Chemistry and Chemical Engineering, Shaanxi Normal University, Xi’an 710119, China; zhangteacher@snnu.edu.cn

**Keywords:** energy storage, metal–insulator–metal capacitors, atomic layer deposition, laser direct writing, electrical performance

## Abstract

This study presents the construction and dielectric properties investigation of atomic-layer-deposition Al_2_O_3_/TiO_2_/HfO_2_ dielectric-film-based metal–insulator–metal (MIM) capacitors. The influence of the dielectric layer material and thickness on the performance of MIM capacitors are also systematically investigated. The morphology and surface roughness of dielectric films for different materials and thicknesses are analyzed via atomic force microscopy (AFM). Among them, the 25 nm Al_2_O_3_-based dielectric capacitor exhibits superior comprehensive electrical performance, including a high capacitance density of 7.89 fF·µm^−2^, desirable breakdown voltage and leakage current of about 12 V and 1.4 × 10^−10^ A·cm^−2^, and quadratic voltage coefficient of 303.6 ppm·V^−2^. Simultaneously, the fabricated capacitor indicates desirable stability in terms of frequency and bias voltage (at 1 MHz), with the corresponding slight capacitance density variation of about 0.52 fF·µm^−2^ and 0.25 fF·µm^−2^. Furthermore, the mechanism of the variation in capacitance density and leakage current might be attributed to the Poole–Frenkel emission and charge-trapping effect of the high-*k* materials. All these results indicate potential applications in integrated passive devices.

## 1. Introduction

As an increasing trend for next-generation radio-frequency (RF) and analog mixed-signal integrated circuits (ICs) applications, a higher level of energy storage and integration performance are required for on-chip system components to reduce chip size and fabrication cost. Metal–insulator–metal (MIM) dielectric capacitors (DCs), as one of the crucial and typical components, have been widely used in silicon integrated RF and ICs devices due to their low resistance and low parasitic capacitance [1,2,3,4,5,6,7,8]. In recent years, a great number of research studies have been devoted to developing energy and capacitance densities in energy storage fields, especially for supercapacitors (SCs) [9,10,11]. Nonetheless, the intrinsic characteristics of SCs, such as low operation voltage and poor working frequency, have greatly limited their electronic applications [12,13,14]. Compared with SCs, DCs adopt dielectric materials sandwiched between the top and bottom electrodes to achieve higher working frequency and voltage due to their fast-moving charges and high-power discharge, which can achieve MHz and exceed 5 V [15,16], respectively. The high capacitance density of MIM capacitors can be realized by using high-*k* dielectric materials or with a lower film thickness [17,18]. Yu et al. investigated a high-density (13 fF·µm^−2^) MIM capacitor using HfO_2_ dielectrics through atomic layer deposition (ALD) [19]. Ding and co-workers reported the use of Al_2_O_3_ dielectric film (13 nm thickness) as the insulator of a dielectric capacitor with a capacitance density of 6.05 fF·µm^−2^, intended for RF and mixed-signal integrated circuits applications [20]. Boris’s group focused on the dynamic random access memory application of a capacitor with TiO_2_-based high-*k* dielectrics [21]. Due to their increased integration application, most previously reported works focused on how to stack or laminate different kinds of high-*k* dielectric materials to achieve high capacitance density and other critical parameters, such as low leakage characteristic, good voltage linearity, and reliability. For instance, Wu et al. utilized the stack composition of TiO_2_ and SiO_2_ to achieve desirable performance in terms of a capacitance density of 11.9 fF·µm^−2^ with a *VCC*-*α* of 90 ppm·V^−2^ [22]. Additionally, Sung and co-workers proposed a laminated Al_2_O_3_/HfO_2_/Al_2_O_3_ MIM capacitor to estimate the leakage current and voltage linearity [23].

In these studies, HfO_2_, Al_2_O_3_, and TiO_2_ are regarded as the representative materials for fabricating MIM capacitors due to their high-*k* and relatively large bandgap energy. However, one of the great challenges is the constraint trade-off between high capacitance density and breakdown field strength because of the direct effect of the dielectric layer thickness on the leakage current and breakdown voltage [24,25]. Herein, we systematically study the synergistic effect of different dielectric material types and their thicknesses on the performance of DCs. Moreover, the physical morphology features and electrical characteristics of the DCs are further investigated. It is found that the Al_2_O_3_-based dielectric capacitor exhibits superior comprehensive electrical performance compared to the DCs based on HfO_2_ and TiO_2_. Moreover, the thickness of the Al_2_O_3_-based dielectric capacitor is further investigated. Accordingly, the optimized Al_2_O_3_-based dielectric capacitor with a thickness of 25 nm demonstrates a lower leakage current, a larger breakdown voltage, a stable frequency and voltage retention, and an ideal secondary voltage coefficient, providing new insights for silicon integrated RF and ICs devices.

## 2. Experimental Section

The basic information of the selected silicon substrate for the experiment includes crystal orientation (<100> = 0.5°), diameter (100 ± 0.2 mm), and thickness (500 ± 10 μm). Aluminum (A1 13503) used in vacuum thermal evaporation process is a specimen (∅3 × 3 mm) with a purity of 99.999%. Firstly, a layer of photoresist (PR) was spin-coated on the silicon substrate, and the bottom electrode was patterned by laser direct writing technology with a four-axis laser-pattern direct writing system (see Appendix A for scalable electrode pattern design, Supporting Information); the final bottom electrode patterned structure was fabricated by the subsequent development process. Subsequently, about 100 nm-thick metal Al was deposited by vacuum thermal evaporation process using a *JSD300* vacuum thermal evaporation coating system as a bottom electrode layer, where Al was employed to enhance conductivity while reducing the parasitic resistance of the electrode. During the entire ALD deposition process, the remaining photoresist could be decomposed spontaneously in a high-temperature environment. The dielectric layers, Al_2_O_3_, TiO_2_, and HfO_2_, were deposited by ALD with the equipment of an *NCE*-*200R* atomic layer deposition system at different temperature conditions. In detail, nitrogen gas was used as a carrier to feed the reaction sources Al(CH_3_)_3_/H_2_O, TiCl_4_/H_2_O and HfCl_4_/H_2_O precursors into the reaction chamber, and the temperature of the reaction chamber was 200 °C, 60 °C, and 100 °C, respectively. Additionally, the corresponding reaction rate was 0.667 nm per cycle for Al_2_O_3_, 0.031 nm per cycle for TiO_2,_ and 0.125 nm per cycle for HfO_2_, respectively. Two different dielectric types (Al_2_O_3_ and TiO_2_) of MIM capacitors with thicknesses of 12.5 nm, 25 nm, and 50 nm were fabricated; simultaneously, HfO_2_ MIM capacitors with thicknesses of 12.5 nm and 25 nm were also prepared for comparisons. Then, the photoresist was secondary spin-coated on each dielectric layer, and the top electrode was patterned again using laser direct writing technology. Finally, Al of 150 nm was prepared as the top electrode with the same method described above for depositing the bottom electrode.

In the experiment, the electrode patterning equipment used a four-axis laser-pattern direct writing system (*Microlab*), the top and bottom electrode evaporation equipment was a *JSD300* vacuum thermal evaporation coating system, and the dielectric layer deposition equipment was employed by an *NCE*-*200R* atomic layer deposition system. The morphology and roughness of the MIM capacitors were characterized by a *Carl Zeiss SIGMA HD* scanning electron microscope (SEM) and a *Dimension Icon* atomic force microscope (AFM). Insulator thicknesses of Al_2_O_3_, TiO_2,_ and HfO_2_ were evaluated via an *SE*-*VE* spectroscopic ellipsometer. The entire parameter test of the samples was carried out on the low-temperature vacuum probe station, where the capacitance-voltage (*C*-*V*) and capacitance-frequency (*C*-*Q*) characteristics were measured using an *Agilent 4294A* semi-conductor parameter analyzer, and leakage current-voltage (*J*-*V*) characteristic measurements were carried out with a *Keysight B2912A* parameter tester.

## 3. Results and Discussion

The planar area of the MIM capacitors is designed to be 80 µm × 80 µm on an n-type single-side polished (100) silicon wafer with a resistivity of 0.01 Ω·cm, and the fabrication process is schematically illustrated in Figure 1, where the detailed preparation process corresponds to the above-mentioned fabrication of MIM silicon capacitors. Moreover, combined with micro/nanofabrication technology, the actual size can be adjusted larger according to different practical applications of the dielectric capacitors, which enables the possibility of mass production. Different from other deposition technologies for dielectric layers, such as electron beam deposition, electrodeposition, and magnetron sputtering, ALD can prepare highly pure and dense thin film, accurately controlling the thickness and composition of the required dielectric layer through reaction conditions simultaneously.

Figure 2a,b shows the SEM and cross-sectional SEM images of the fabricated MIM capacitors with a planar area of 80 µm × 80 µm. In detail, 100 nm and 150 nm Al work as the top/bottom electrode, which are uniformly evaporated by the vacuum thermal evaporation process, and 25 nm Al_2_O_3_ deposited by ALD acts as the dielectric layer. As can be seen, the manufactured MIM capacitors exhibit a step effect with the patterning of the top and bottom electrodes and the uniformity of the vacuum thermal evaporation coating in Figure 2a. Figure 2b shows a clear boundary at the interface between different layers, indicating the outstanding advantage of the compact high-*k* dielectric layers by low-temperature ALD deposition, film composition control, and outstanding semi-conductor compatibility [26]. Moreover, the prepared low-temperature ALD dielectric layers are amorphous, which can reduce the leakage current along the edge of the grain boundaries [27]. The dielectric capacitors based on Al_2_O_3_, TiO_2_, and HfO_2_ with controllable thicknesses are successfully fabricated by ALD, and the thicknesses of the dielectric film are measured with a spectroscopic ellipsometer. The relevant test principle is shown in Appendix A. Figure 2c proves that the curve fitted by the model is consistent with the experimental test curve, indicating the accuracy of the test data. The relationship between the refractive index *n_2_* of the film layer to be measured and the thickness *d* can be obtained. Figure 3d shows the XPS survey spectrum to determine the chemical composition of the dielectric layer Al_2_O_3_ prepared by ALD. The XPS survey spectra represent mainly Al, O, and C contributions. The XPS measurement spectrum shows that typical peaks appear at 74 eV, 118 eV, 285 eV, 532 eV, 979 eV, and 1230 eV, representing the binding energies of Al 2p, Al 2s, and C 1s, O 1s, O KLL, and C KLL, respectively [28]. The peaks corresponding to the plasmon losses are also observed [29]. As can be seen from Figure 3d, the bulk plasmon loss peaks are detected at approximately 554 and 153 eV, corresponding to bulk plasmon energy $E_{P}=554\;\mathrm{eV}-E_{O\;1s}=22\;\mathrm{eV}$ and $E_{P}=153\;\mathrm{eV}-E_{\mathrm{Al}\;2s}=35\;\mathrm{eV}$. High-resolution XPS spectra of the Al 2p and O 1s are shown in Figure 2e,f. Al 2p peak could be fitted as two asymmetric single peaks. The presence of Al-O and Al-OH bonds in the Al_2_O_3_ film is confirmed by the binding energies of the Al 2p peak at 74.7 eV and 75.0 eV. The O 1s peak at 532.0 eV is relatively broad and asymmetric as it is associated with four types of bonds (Figure 2f). Further deconvolution revealed four/three distinct components, the strongest peak located at 531.2 eV originated from Al-O bonds, and the other peak at 532.0 eV associated with Al-O-H hydroxyl groups appeared because of the water-containing raw materials for Al_2_O_3_ film growth due to ALD [30]. The peak at 532.9 eV for the Al_2_O_3_ film is due to C=O radicals. The peak at 530.5 eV for the Al_2_O_3_ film could be related to adsorbed oxygen [31].

To further verify the morphology and roughness of the dielectric layer after ALD deposition, the as-prepared samples with different dielectric materials and thicknesses are measured by AFM. Figure 3a–c shows the topographies of Al_2_O_3_, TiO_2_, and HfO_2_ film with a thickness of 25 nm-based MIM capacitors. The analysis result reflects that the root-mean-square (*Rq*) roughness values of these three materials are only 2.18 nm, 2.57 nm, and 2.22 nm, which can indirectly depict the special surface area [32,33,34]. It was noticed that there are certain white “defects” in the film itself after ALD deposition, and these white “defects” actually refer to the peaks of the surface undulation of the dielectric layer. The defects of the TiO_2_ film in Figure 3b are more pronounced, indicating that the film quality of TiO_2_ is inferior to that of Al_2_O_3_ and HfO_2_, which is mainly due to a lower deposition temperature. Figure 3d shows the linear contour fluctuations in the directions indicated by the black, red, and blue lines in Figure 3a–c. It can be seen that the linear fluctuation of TiO_2_ is the largest, while the profile fluctuations of Al_2_O_3_ and HfO_2_ are relatively more stable. Additionally, it can be concluded that the 25 nm dielectric films deposited by ALD display better uniformity and compactness compared to the 12.5 nm and 50 nm deposition (Appendix A).

Figure 4 indicates the typical *J*-*V* characteristics of the leakage current density and breakdown voltage under positive and negative voltages. In general, the breakdown strength of the MIM capacitor reflects the device lifetime [35]. Therefore, a continuously increasing voltage is applied to the capacitor until electrical breakdown occurs. With the increase in the voltage, the resulting high electric field broke down the layer of dielectric capacitors. Take the diagram of Al_2_O_3_ dielectric capacitors in Figure 4a as an example for detailed analysis, it is found that the MIM capacitor with a 12.5 nm Al_2_O_3_ dielectric structure has a low leakage current density of about 5.3 × 10^−9^ A·cm^−2^ at 4.3 V, which meets the requirement of high-density capacitor applications [36]. Simultaneously, the structures of 25 nm and 50 nm capacitors were measured for high breakdown voltages rising from 12 V to 21.6 V, and the corresponding leakage current densities up to 1.4 × 10^−10^ A·cm^−2^ and 2.4 × 10^−11^ A·cm^−2^, respectively. High capacitance density is one of the vital indicators for evaluating the quality of capacitors, and its variations are mainly caused by the charge-trapping effect between the electrode and the surface of the dielectric [37,38]. However, the increase in the breakdown voltage is largely at the expense of capacitance density. Therefore, in different applications (power supply bypass, high-density, high-precision capacitors), the selection of the optimal parameters should be considered comprehensively.

The *J*-*V* curve can be divided into two regions at 4.6 V, where the leakage current densities increase sharply with the bias voltage (breakdown voltage). This can be explained by the different conductive mechanisms of leakage current, namely Schottky emission and Poole–Frenkel emission at low and high electric fields [5,36,39], which are shown by (1) and (2), respectively.
(1)$$J={AT}^{2}\;exp\;[-(q\emptyset_{S}-\beta_{S}E^{1/2})/kT]$$
(2)$$J=CE\;exp\;[-(q\emptyset_{PF}-\beta_{PF}E^{1/2})/kT]$$
where *A* and *C* are constants; *T* is the temperature in kelvin, 298 K; *E* is the electric field; *E = U/d*; *q* is the electron charge; ∅*_S_* is the barrier height of the interface between the dielectric and injecting electrode for Schottky emission; ∅*_PF_* is the trap height in the dielectric for *PF* emission; *k* is the Boltzmann constant; *β_S_* and *β_PF_* are (*q*^3^/*πε*_0_*n*^2^)^1/2^ and (*q*^3^/*πε*_0_*n*^2^)^1/2^, respectively, in which *ε*_0_ is a permittivity in a vacuum; and *n* is the refractive index.

Equations (1) and (2) are fitted to the experimental data and extract the *n* value of the refractive index from the slope of the fitted curve, which further indicates that the conduction mechanism of the MIM capacitor is dominated by PF emission at high electric region. In fact, the neutral electron traps in high-*k* materials are abundantly generated under an applied electric field, resulting in an increase in leakage current. Hence, the leakage current is dominated by the trap-assisted-tunneling in the low electric field region. As can be seen, the leakage currents decrease with the increasing dielectric thickness. On the contrary, in the case of the same film thickness, the increase in the bias voltage proves that the traps still exist within the films prepared by ALD, which is possibly related to the parameters of the preparation process, including temperature, vacuum, and substrate crystallinity [40]. In particular, the breakdown voltages and leakage current densities for TiO_2_ of 13.52 nm, 26.10 nm, and 52.54 nm are 0.44 V and 2.3 × 10^−8^ A·cm^−2^, 0.58 V and 1.7 × 10^−8^ A·cm^−2^, and 0.85 V and 1.4 × 10^−9^ A·cm^−2^. Moreover, for HfO_2_, the breakdown voltages and leakage current densities of 10.51 nm and 21.29 nm-thick dielectric capacitors are 0.8 V and 2.4 × 10^−9^ A·cm^−2^ and 9.9 V and 8.4 × 10^−10^ A·cm^−2^, respectively. From the perspective of dielectric materials, the *J*-*V* curves indicate that Al_2_O_3_ holds better withstand voltage characteristics, which is possibly due to the superior compactness of dielectric film fabricated at 200 °C.

Figure 5a–c demonstrates the dependence of capacitance density as a function of frequency. As can be seen, the capacitance density of Al_2_O_3_ MIM capacitors with different thicknesses (12.10 nm, 24.72 nm, and 49.73 nm) exhibit a slight degradation from 0 to 1 MHz, and the maximum change values are only 1.45, 0.54, and 0.17 fF·µm^−2^, which reflects the excellent dielectric characteristic of Al_2_O_3_ MIM capacitors with frequencies. Nevertheless, under the same preparation conditions, the capacitance densities of TiO_2_ vary in the range of 5.70, 3.48, and 3.20 fF·µm^−2^, respectively. For HfO_2_, the corresponding values are 2.51 and 2.03 fF·µm^−2^. Compared with Al_2_O_3_ dielectric films, neither TiO_2_ nor HfO_2_ are relatively satisfactory results. Thus, it can be concluded that the large bandgap energy (8.9 eV) of the larger barrier type Al_2_O_3_ brings excellent frequency stability, while the bandgap energies of TiO_2_ and HfO_2_ are only 3.2 eV and 5.6 eV, respectively [23]. The large bandgap leads to a higher energy requirement of the electrons in the semi-conductor material when the energy level transition occurs, thereby providing a relatively stable capacitance density capability for the MIM capacitor. As can be seen from Figure 5b,c, there are a large number of charge traps near the interface between electrodes and dielectric layers, and the electrons still become inactive when the frequency increases [41]. Therefore, the high bandgap barrier capacitor has fewer charge traps near the surface of the dielectric layer, or these traps make the capacitance density only function in the low-frequency range [42,43]. In either case, the above two situations are conducive to maintaining the capacitance density of Al_2_O_3_ MIM capacitors stable with frequency. As confirmed in Figure 5a–c, the capacitance densities for 12.10 nm, 24.72 nm, and 49.73 nm Al_2_O_3_ capacitors are 13.19 fF·µm^−2^, 7.89 fF·µm^−2^, and 4.62 fF·µm^−2^, respectively.

Figure 5d–f reflects the variation of capacitance with applied voltage for Al_2_O_3_ MIM capacitors at different frequencies. It can be inferred from Figure 5d,e that the overall capacitance density of the 12.5 nm-thickness Al_2_O_3_ dielectric capacitor decreases slightly with the bias voltages, and the 25 nm-thickness capacitor increases slightly, whereas the increase and decrease in the maximum values are only 1.19 fF·µm^−2^ and 0.25 fF·µm^−2^. The reason for the different variation trends may be related to the thickness of the amorphous structures prepared by low-temperature ALD deposition. As is well-known, capacitance density is significantly sensitive to dielectric (Al_2_O_3_) thickness due to the relatively low dielectric constant of Al_2_O_3_. It can be seen from Figure 5d–f that the 12.5 nm and 25 nm-thick Al_2_O_3_ MIM capacitors show a slight change with voltage, while the 50 nm-thick Al_2_O_3_ MIM capacitors remain almost unchanged at the same voltage bias. In general, the capacitance densities of Al_2_O_3_ capacitors change by 0.45% under the frequency of 10^3^ Hz, 10^4^ Hz, 10^5^ Hz, and 10^6^ Hz, indicating exceptional stability. At the same time, they show higher capacitance characteristics under the low-frequency region, which is related to the charge trapping effect on the surface of the electrode and the dielectric layer. That is, when a bias voltage is applied to the electrode, the traps in the dielectric layer cause the charge trapping of the carriers, leading to the change of capacitance with frequency [44].

Previously, we measured and analyzed the *C*-*Q* and *C*-*V* characteristic curves of MIM capacitors with three thicknesses represented by Al_2_O_3_. Combined with *C*-*Q* and *C*-*V* curves, a higher dielectric layer thickness has better stability in terms of bias voltage and frequency. Nonetheless, the obtained capacitance density will be reduced, due to the working principle of planar capacitors, and the conclusion is in line with our expectations. For this reason, taking a 25 nm dielectric layer thickness as an example, the variation trends of three kinds of dielectric materials with increasing bias voltage and frequency are also discussed. Figure 6a–c reflects the dependence of capacitance density with Al_2_O_3_, TiO_2_, and HfO_2_ dielectric materials as a function of frequency. It is observed that the capacity densities of the three dielectric capacitors decrease slightly with frequency, but the values do not change much. Among them, the maximum change rate for Al_2_O_3_, TiO_2_, and HfO_2_ is only 0.5 fF·µm^−2^, 3.5 fF·µm^−2^, and 1.9 fF·µm^−2^, respectively. At the same time, the capacitance density of the TiO_2_ dielectric films has large fluctuations with frequency, which is related to a large number of defects on the surface of TiO_2_ films prepared by ALD deposition under 60 °C and vacuum conditions. The calculated average capacitance densities are 7.89 fF·µm^−2^ (24.72 nm Al_2_O_3_), 21.08 fF·µm^−2^ (26.20 nm TiO_2_), and 12.29 fF·µm^−2^ (21.29 nm HfO_2_). Figure 6d–f exhibits the voltage stability with different dielectric materials. It should be pointed out that the capacitance density of TiO_2_ has a relatively large fluctuation at 1 kHz, which further proves the structure morphology’s direct impact on the electrochemical performance of the MIM capacitors.

The voltage coefficients of capacitance (VCCs) are one of the most important parameters to evaluate the voltage linearity of the MIM capacitor, and it is the key factor to consider in the application of bypass power supply and the radio frequency circuit, which can be determined by an equation [45] and combined with experimental data to fit VCCs. *C* (*V*) = *C*_0_ (*αV*^2^ + *βV* + 1) → (*C* (*V*) *− C*_0_)/*C*_0_ = [*αV*^2^ + *βV*]*_ppm_*, where *C*_0_ represents the zero-bias capacitance, *α* and *β* are the quadratic and linear voltage coefficients, respectively. Among them, the quadratic voltage coefficient *α* is the most critical parameter for evaluating voltage linearity, where *β* can be eliminated by circuit design or canceled out by differential methods [18,46,47]. Figure 7 indicates the normalized capacitance of the 25 nm Al_2_O_3_ MIM capacitors with bias voltages of 10^4^ Hz, 10^5^ Hz, and 10^6^ Hz. It can be seen that as the frequency increases, the normalized capacitance slope shows a downward trend, while the quadratic voltage coefficient *α* values decrease from 329.1 ppm·V^−2^ to 270.5 ppm·V^−2^, which is attributed to the increase in the relaxation time and the smaller capacitance variation caused by the gradually increasing high frequency [48].

## 4. Conclusions

In this work, three types of high-performance MIM dielectric capacitors (Al_2_O_3_, TiO_2_, and HfO_2_, with an area of 80 µm × 80 µm) on a silicon substrate are successfully fabricated by the combination of vacuum thermal evaporation, laser direct writing technology, and atomic layer deposition (ALD). The physical morphology and electrochemical properties of the as-prepared MIM dielectric capacitors are also systematically investigated. The prepared samples show a certain discrepancy in compactness and surface defects. Among them, the Al_2_O_3_ and HfO_2_ insulators demonstrate better compactness and fewer surface defects, while the TiO_2_ insulator shows a relatively poor preparation effect, which is related to the fabrication process and substrate defects. Through the precisely controlled micro-machining processes, the manufactured 25 nm Al_2_O_3_ capacitor exhibits a capacitance density up to 7.89 fF·µm^−2^, a leakage current lower than 1.4 × 10^−10^ A·cm^−2^, a breakdown voltage of 12 V, and a breakdown electric field of 4.8 MV·cm^−1^, which can satisfy the requirements of integrated passive devices for capacitors. Based on the restrictive trade-off between leakage current and capacitance density, the relationship between dielectric layer material and thickness is further discussed. Three-dimensional (3D) multilayer silicon-based capacitors are believed to achieve a larger capacitance density, and different 3D structures may have a great influence on the current distribution and the performance effect of MIM capacitors, which are directions worth studying in the future.

## Figures and Tables

**Figure 1 molecules-27-03951-f001:**
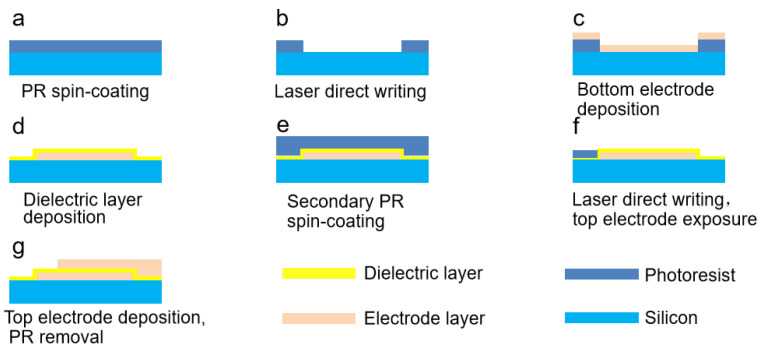
Schematic illustration of fabrication process of MIM capacitors. (**a**) PR spin-coating. (**b**) Laser direct writing. (**c**) Bottom electrode deposition by laser direct writing. (**d**) Dielectric layer deposition by ALD. (**e**) Secondary PR spin-coating. (**f**) Laser direct writing and top electrode exposure. (**g**) Top electrode deposition and PR removal.

**Figure 2 molecules-27-03951-f002:**
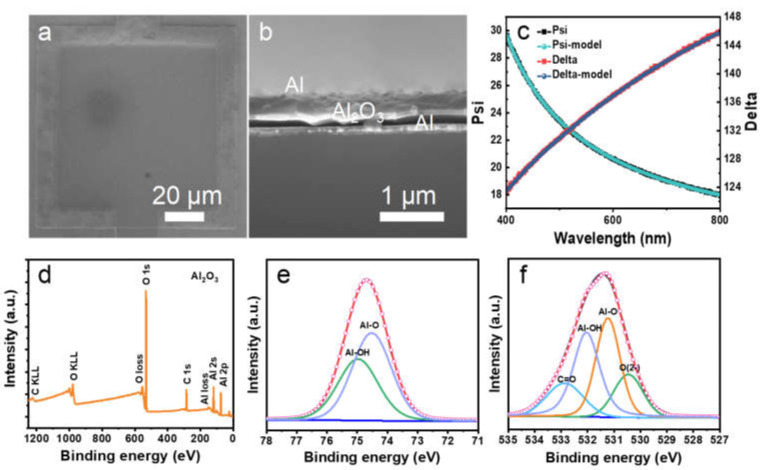
(**a**) SEM image of the Al_2_O_3_-based MIM capacitor. (**b**) The cross-sectional SEM image of the MIM capacitor. (**c**) The 25 nm Al_2_O_3_ film thickness test results obtained by ellipsometry spectrometer: where Psi represents the amplitude ratio, and Delta represents the phase difference. (**d**) XPS survey spectra of the prepared Al_2_O_3_ film. High-resolution XPS spectra of (**e**) Al 2p, (**f**) O 1s.

**Figure 3 molecules-27-03951-f003:**
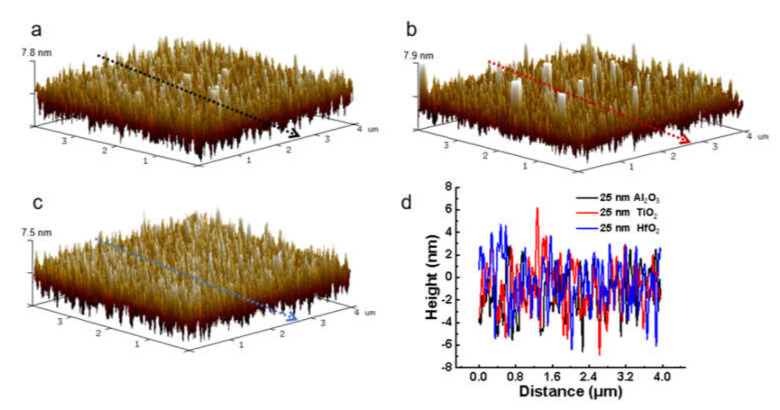
AFM images of MIM capacitors deposited with 25 nm Al_2_O_3_, TiO_2_, and HfO_2_ dielectric material, respectively. (**a**) ALD deposition of Al_2_O_3_ dielectric material. (**b**) ALD deposition of TiO_2_ dielectric material. (**c**) ALD deposition of HfO_2_ dielectric material. (**d**) The linear contour fluctuations in the directions are indicated by the black, red, and blue lines in Figure 3a–c.

**Figure 4 molecules-27-03951-f004:**
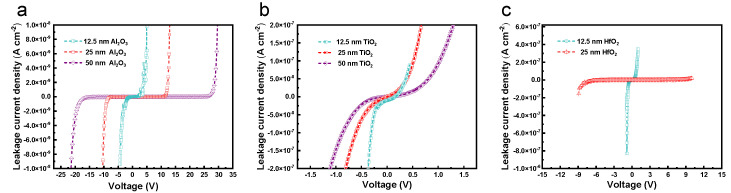
The relationship between leakage current density and applied voltage of manufactured MIM capacitors with different dielectric materials and thicknesses. (**a**) *J*-*V* characteristics of Al_2_O_3_ dielectric MIM capacitors. (**b**) *J*-*V* characteristics of TiO_2_ dielectric MIM capacitors. (**c**) *J*-*V* characteristics of HfO_2_ dielectric MIM capacitors.

**Figure 5 molecules-27-03951-f005:**
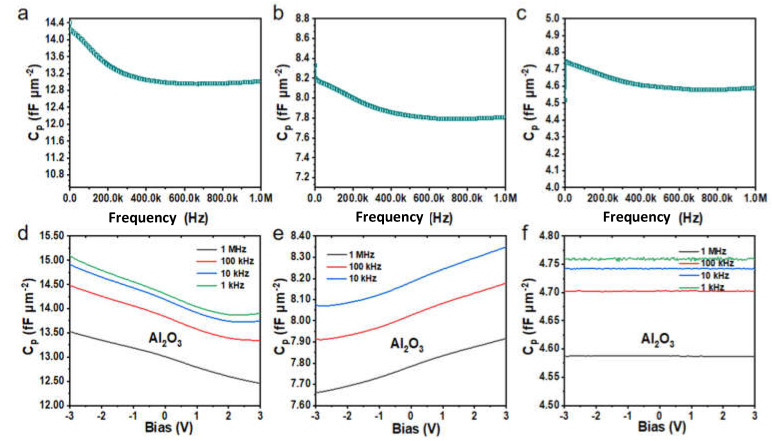
*C*-*Q* and *C*-*V* characteristic curves of 12.10 nm, 24.72 nm, and 49.73 nm Al_2_O_3_ dielectric capacitors. (**a**–**c**) The *C*-*Q* characteristic curves of three thicknesses of dielectric capacitors, respectively. (**d**–**f**) The corresponding *C*-*V* characteristic curves of three thicknesses of dielectric capacitors at different frequencies.

**Figure 6 molecules-27-03951-f006:**
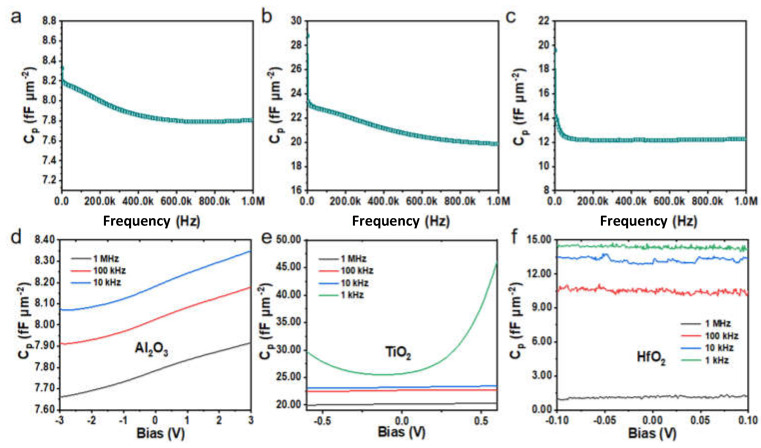
*C*-*Q* and *C*-*V* characteristic curves of 25 nm Al_2_O_3_, TiO_2_, and HfO_2_ dielectric capacitors. (**a**–**c**) The *C*-*Q* characteristic curves of three kinds of 25 nm dielectric capacitors, respectively. (**d**–**f**) The corresponding *C*-*V* characteristic curves of three kinds of 25 nm dielectric capacitors at different frequencies.

**Figure 7 molecules-27-03951-f007:**
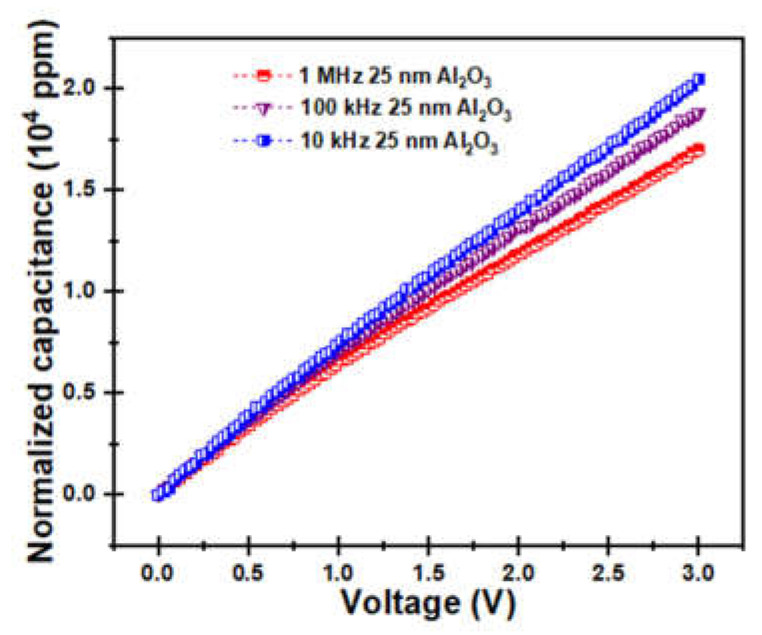
Normalized capacitance as the function of bias voltage of 25 nm Al_2_O_3_ capacitors at 10^4^ Hz, 10^5^ Hz, and 10^6^ Hz, respectively.

## Data Availability

The data presented in this study are available in the article.

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
