# Peer review of "Dielectric Properties Investigation of Metal–Insulator–Metal (MIM) Capacitors"

_molecules, 2022, doi:10.3390/molecules27123951_

Round 1

Reviewer 1 Report

The article of Xiong et al. reports on the fabrication and dielectric properties characterization of three different MIM dielectric capacitors on silicon substrate. For this, they used the following dielectric materials: Al2O3, TiO2, and HfO2 that were grown by atomic layer deposition technique. For dielectric properties, C-V, C-Q and J-V characteristics were analyzed. In general the paper is very well written and the results are interesting for the readers of the Molecules MDPI journal. In order to improve text quality some corrections are suggested:

1. Abstract: Please verify at line 19 the use of "@" as "and": "12 [email protected] 10-10 A·cm-2 ,";

2. Experimental section:

(a) Concern the temperature values of the ALD process "200 ℃, 60 ℃, and 100 ℃", it is not clear if each value is for on given material, please improve this phrase in the text;

(b) Please inform the number of reaction cycles for each metal oxide layer, as well as the purge gas during ALD process.

3. Results and discussion:

(a) Page 5, line 208-210: Also verify the use of @.

Author Response

Responses to Reviewers

Reviewer #1: The article of Xiong et al. reports on the fabrication and dielectric properties characterization of three different MIM dielectric capacitors on silicon substrate. For this, they used the following dielectric materials: Al2O3, TiO2, and HfO2 that were grown by atomic layer deposition technique. For dielectric properties, C-V, C-Q and J-V characteristics were analyzed. In general the paper is very well written and the results are interesting for the readers of the Molecules MDPI journal. In order to improve text quality some corrections are suggested:

Response: We acknowledge the reviewer for his/her enthusiastic and positive review of our work. We also thank him/her for bringing up new comments, which help us improve our results. In the following, we reply one by one to the referee’s questions.

  1. Abstract: Please verify at line 19 the use of "@" as "and": "12 [email protected] 10-10 A·cm-2 ,";

Response: We thank the reviewer’s professional suggestions and comments, which make our paper better. In the revised version of the manuscript, the “@” mentioned in the text is replaced by “and”. The sentence “Among them, 25 nm Al2O3-based dielectric capacitor exhibits superior comprehensive electrical performance, including a high capacitance density of 7.89 fF·µm-2, the desirable breakdown voltage and leakage current of about 12 V and 1.410-10 A·cm-2, and the quadratic voltage coefficient of 303.6 ppm·V-2.” was added on page 1.

  1. Experimental section:

(a) Concern the temperature values of the ALD process “200 ℃, 60 ℃, and 100 ℃”, it is not clear if each value is for on given material, please improve this phrase in the text;

(b) Please inform the number of reaction cycles for each metal oxide layer, as well as the purge gas during ALD process.

Response: We sincerely thank the reviewer for the valuable suggestions and comments. In the revised version of the manuscript, the temperature, gas and growth rate during ALD deposition are listed to make the whole experiment process clearer. The sentence “The dielectric layers Al2O3, TiO2 and HfO2 were deposited by ALD with the equipment of a NCE-200R atomic layer deposition system at different temperature conditions. In detail, nitrogen gas was used as a carrier to feed the reaction sources Al(CH3)3/H2O, TiCl4/H2O and HfCl4/H2O precursors into the reaction chamber, and the temperature of the reaction chamber was 200 ℃, 60 ℃ and 100 ℃, respectively. And the corresponding reaction rate was 0.667 nm per cycle for Al2O3, 0.031 nm per cycle for TiO2 and 0.125 nm per cycle for HfO2, respectively.” was added on page 4.

  1. Results and discussion: (a) Page 5, line 208-210: Also verify the use of @.

Response: We really appreciate the helpful suggestion. In the revised version of the manuscript, the “@” mentioned in the text is replaced by “and”. The sentences “In particular, the breakdown voltages and leakage current densities for TiO2 with 13.52 nm, 26.10 nm and 52.54 nm are 0.44 V and 2.310-8 A·cm-2, 0.58 V and 1.710-8 A·cm-2, 0.85 V and 1.410-9 A·cm-2. Besides, for HfO2, the breakdown voltages and leakage current densities of 10.51 nm and 21.29 nm thick dielectric capacitors are 0.8 V and 2.410-9 A·cm-2, 9.9 V and 8.410-10 A·cm-2, respectively. ” were added on page 9.

Reviewer 2 Report

The paper presents a dielectric properties investigation of metal-insulator-metal (MIM) capacitors. According to the reviewer’s opinion, the paper is well-structured and clear. The topic is interesting and falls within the aim of the journal. In addition, the results are well-presented and could be helpful to further develop the same topic. Therefore, the paper can be accepted for publication in the current form.

Author Response

Reviewer #2: The paper presents a dielectric properties investigation of metal-insulator-metal (MIM) capacitors. According to the reviewer’s opinion, the paper is well-structured and clear. The topic is interesting and falls within the aim of the journal. In addition, the results are well-presented and could be helpful to further develop the same topic. Therefore, the paper can be accepted for publication in the current form.

Response: We highly appreciate the reviewer’s positive comments on our work.

Reviewer 3 Report

 In this paper, the authors present the fabrication and the dielectric properties of metal-insulator-metal (MIM) capacitors with Al2O3, TiO2, or HfO2 dielectrics obtained by atomic-layer-deposition. They characterize the dielectric films using ellipsometry, XPS, and AFM and systematically study the influence of dielectric material and layer thickness on the performance of MIM capacitors. They find that the Al2O3-based dielectric capacitor exhibits superior electrical performance over HfO2 and TiO2.  

 Although of limited novelty, the study is interesting and can serve for the potential application in integrated passive devices. The manuscript is rather clear and well organized. The results and the related discussion are convincing.

 The paper can be suitable for publication after a revision. Here are some points that requires attention:

 1.     In the introduction: “ In recent years, a great number of researches have been devoted to developing energy and capacitance densities in energy storage fields, especially for supercapacitors (SCs) [9, 10].” I suggest adding this recent paper on supercapacitors:  https://doi.org/10.1002/admi.202100149

 2.     “Besides, the prepared low-temperature ALD dielectric layers are amorphous, which can  reduce the leakage of leakage current along the edge of the grain boundaries [26]” I guess there is a typo in this sentence.

3.     “As it was noticed that there are certain white defects in the film itself after ALD deposition, which are mainly due to the compatibility of different dielectric layers with process parameters during deposition.” Can the authors better define these defects and explain their origin? Besides these white defects seem not to be present in the HfO2 dielectric material.

4.     “The J-V curve can be divided into two regions at 4.6 V, where the leakage current densities increase sharply with the bias voltage (breakdown voltage). This can be explained by different conductive mechanisms, namely Poole-Frenkel emission and Schottky emission at low and high electric fields [5, 35, 38].” Poole-Frenkel and Schottky emission are only mentioned here but highlighted in the abstract. The authors should elaborate more on this point and possible show fits of the JV dependencies according to the recalled mechanisms.

5.     “As can be seen, the capacitance density of Al2O3 MIM capacitors with different thicknesses (12.10 nm, 24.72 nm and 49.73 nm) exhibit a slight degradation from 0 to 1 221 MHz, and the values only range from 0.07 to 1.38 fF·μm-2, which reflects the excellent dielectric characteristic with frequencies.”  The quoted values 0.07 to 1.38 fF·μm-2 do not seem to match what is reported in plots 5a-c or it is not clear what they are.

6.     “This can be concluded that the large band gap energy of barrier type Al2O3 226 (8.9 eV), while the band gap energies of TiO2 and HfO2 are 3.2 eV and 5.6 eV [22].” I am not sure that I understand the meaning of this sentence which seems to me incomplete. Please rephrase it.

7.     “It can be inferred from Fig. 5d and Fig. 5e that the overall capacitance density of the 12.5 nm thickness Al2O3 dielectric capacitor decreases slightly with the bias voltages, and the 25 nm thickness capacitor increases slightly, whereas the increase and decrease of the maximum values are only 1.19 fF·μm-2 and 0.25 fF·μm-2.” Have the authors any explanation of this different trend of the capacitance vs bias?

Author Response

Reviewer #3: In this paper, the authors present the fabrication and the dielectric properties of metal-insulator-metal (MIM) capacitors with Al2O3, TiO2, or HfO2 dielectrics obtained by atomic-layer-deposition. They characterize the dielectric films using ellipsometry, XPS, and AFM and systematically study the influence of dielectric material and layer thickness on the performance of MIM capacitors. They find that the Al2O3-based dielectric capacitor exhibits superior electrical performance over HfO2 and TiO2

Although of limited novelty, the study is interesting and can serve for the potential application in integrated passive devices. The manuscript is rather clear and well organized. The results and the related discussion are convincing.

The paper can be suitable for publication after a revision. Here are some points that requires attention:

Response: We acknowledge the reviewer for his/her professional evaluation of our work. We also thank him/her for bringing up valuable comments, which help us improve the results. In the following, we reply one by one to the referee’s questions.

  1. In the introduction: “In recent years, a great number of researches have been devoted to developing energy and capacitance densities in energy storage fields, especially for supercapacitors (SCs) [9, 10].” I suggest adding this recent paper on supercapacitors: https://doi.org/10.1002/admi.202100149.

Response: We thank the reviewer’s suggestions and comments, which make our paper better. In the revised version of the manuscript, we added the recommended article as ref.[11]. The sentence “In recent years, a great number of researches have been devoted to developing energy and capacitance densities in energy storage fields, especially for supercapacitors (SCs) [9-11].” was added on page 2.

  1. “Besides, the prepared low-temperature ALD dielectric layers are amorphous, which can reduce the leakage of leakage current along the edge of the grain boundaries [26]” I guess there is a typo in this sentence.

Response: We are grateful for the reviewer’s suggestion. We modified the sentence, “Besides, the prepared low-temperature ALD dielectric layers are amorphous, which can reduce the leakage current along the edge of the grain boundaries [27]. ” on page 10.

  1. “As it was noticed that there are certain white defects in the film itself after ALD deposition, which are mainly due to the compatibility of different dielectric layers with process parameters during deposition.” Can the authors better define these defects and explain their origin? Besides these white defects seem not to be present in the HfO2 dielectric material.

Response: We appreciate the reviewer for the detailed comment. In our original work, we defined the white spot in the AFM image as a “defect”, which might cause misleading, in fact, the white spot in the AFM just reflects a peak in the film topography roughness there. More white spots also reflect the uniformity of the prepared film to a certain extent. Furthermore, the related discussion “As it was noticed that there are certain white “defects” in the film itself after ALD deposition, and these white “defects” actually refer to the peaks of the surface undulation of the dielectric layer. The defects of the TiO2 film in Fig. 3b are more pronounced, indicating that the film quality of TiO2 is inferior to that of Al2O3 and HfO2, which is mainly due to the lower deposition temperature.” was added on page 7 in our revised main text.

We sincerely thank the reviewer for this valuable comment that will motivate us to think deeply and formulate rigorously in our future work.

  1. “The J-V curve can be divided into two regions at 4.6 V, where the leakage current densities increase sharply with the bias voltage (breakdown voltage). This can be explained by different conductive mechanisms, namely Poole-Frenkel emission and Schottky emission at low and high electric fields [5, 35, 38].” Poole-Frenkel and Schottky emission are only mentioned here but highlighted in the abstract. The authors should elaborate more on this point and possible show fits of the JV dependencies according to the recalled mechanisms.

Response: We really appreciate the professional opinion of the reviewer. Indeed, the Poole-Frenkel mechanism and Schottky mechanism involved in the J-V curve are actually generally accepted explanation mechanisms, but we did not explain them in detail, we made further explanation in the text according to the reviewer’s suggestion. However, we do not think it is necessary to further verify the correctness of the mechanism by curve fitting through the Poole-Frenkel mechanism, because similar curves fitting have been done in the literatures [Micromachines. 2018, 9, 69; Solid State Electronics. 2013, 79, 218-222] we cited, and thus the conclusion in our work is convincing. The sentences “The J-V curve can be divided into two regions at 4.6 V, where the leakage current densities increase sharply with the bias voltage (breakdown voltage). This can be explained by different conductive mechanism of leakage current, namely Schottky emission and Poole-Frenkel emission at low and high electric fields [5, 36, 39], which are shown by (1) and (2), respectively.

                                           J=AT2 exp[- (qS - βSE1/2)/kT]                                       (1)

                                          J=CE exp[- (qPF - βPFE1/2)/kT]                                     (2)

Where A and C are constants, T is the temperature in kelvin, 298 K, E is the electric field, E=U/d, q is the electron charge, S is the barrier height of the interface between the dielectric and injecting electrode for Schottky emission, and PF is the trap height in the dielectric for PF emission, k is the Boltzmann constant, βS and βPF are (q3/πε0n2)1/2 and (q3/πε0n2)1/2, respectively, in which ε0 is a permittivity in vacuum, n is the refractive index.

Equations (1) and (2) are fitted to the experimental data and extract the n value of the refractive index from the slope of the fitted curve, which further indicate that the conduction mechanism of the MIM capacitor is dominated by PF emission at high electric region. In fact, the neutral electron traps in high-k materials are abundantly generated under an applied electric field, resulting in an increase in leakage current. Hence, the leakage current is dominated by the trap-assisted-tunneling at low electric field region. ” was added on page 9.

We sincerely thank the reviewer for this valuable comment that will motivate us for in-depth thinking and investigation in our work.

  1. “As can be seen, the capacitance density of Al2O3 MIM capacitors with different thicknesses (12.10 nm, 24.72 nm and 49.73 nm) exhibit a slight degradation from 0 to 1 MHz, and the values only range from 0.07 to 1.38 fF·μm-2, which reflects the excellent dielectric characteristic with frequencies.” The quoted values 0.07 to 1.38 fF·μm-2 do not seem to match what is reported in plots 5a-c or it is not clear what they are.

Response: We are grateful for the reviewer’s suggestion. Indeed, what we originally wanted to express was that Al2O3 MIM capacitors with different thicknesses exhibited a change in capacitance density from 0 to 1 MHz, and the corresponding change value for 12.10 nm Al2O3 is the difference between the value (14.40146 fF·μm-2) at 0 Hz  and the capacitance value (13.01773 fF·μm-2) at 1 MHz, and the result is 1.38 fF·μm-2, With the same calculation method, for 24.72 nm Al2O3, the difference between the value (8.32896 fF·μm-2) at 0 Hz and the capacitance value (7.8101 fF·μm-2) at 1 MHz is 0.51886 fF·μm-2, and for 49.73 nm Al2O3, the difference between the value (4.5897 fF·μm-2) at 0 Hz and the capacitance value (4.51764 fF·μm-2) at 1 MHz is 0.07 fF·μm-2, so the change range is from 0.07-1.38 fF·μm-2. But in fact, this expression is not professional and clear, and as suggested by the reviewer, we give the maximum change value of each capacitance-frequency curve one by one, so that the given values more closely match with Fig. 5d and Fig. 5e. The sentences “As can be seen, the capacitance density of Al2O3 MIM capacitors with different thicknesses (12.10 nm, 24.72 nm and 49.73 nm) exhibit a slight degradation from 0 to 1 MHz, and the maximum change values are only 1.45, 0.54 and 0.17 fF·µm-2, which reflects the excellent dielectric characteristic of Al2O3 MIM capacitors with frequencies. Nevertheless, under the same preparation conditions, the capacitance densities of TiO2 vary in the range of 5.70, 3.48 and 3.20 fF·µm-2, respectively. For HfO2, the corresponding values are 2.51 and 2.03 fF·µm-2. Compared with Al2O3 dielectric films, neither TiO2 nor HfO2 are relatively satisfactory results. ” was added on page 10.

  1. “This can be concluded that the large band gap energy of barrier type Al2O3 (8.9 eV), while the band gap energies of TiO2 and HfO2 are 3.2 eV and 5.6 eV [22].” I am not sure that I understand the meaning of this sentence which seems to me incomplete. Please rephrase it.

Response: We thank the reviewer’s suggestions and comments, which make our paper better. Indeed, this sentence is incomplete and the reader cannot understand the meaning of this sentence, so we revised this sentence. The sentence “This can be concluded that the large band gap energy (8.9 eV) of the larger barrier type Al2O3 brings excellent frequency stability, while the band gap energies of TiO2 and HfO2 are only 3.2 eV and 5.6 eV, respectively [23].” was added on page 10.

  1. “It can be inferred from Fig. 5d and Fig. 5e that the overall capacitance density of the 12.5 nm thickness Al2O3 dielectric capacitor decreases slightly with the bias voltages, and the 25 nm thickness capacitor increases slightly, whereas the increase and decrease of the maximum values are only 1.19 fF·μm-2 and 0.25 fF·μm-2.” Have the authors any explanation of this different trend of the capacitance vs bias?

Response: We really appreciate the helpful comments. In fact, for dielectric capacitors, the relationship of capacitance with bias voltage is

C(V)=C0(αV2+βV+1)

(C(V)-C0)/C0=[αV2+βV]ppm

Where C0 represents the zero-bias capacitance, α and β are the quadratic and linear voltage coefficients, respectively. Among them, quadratic voltage coefficient α is the most critical parameter for evaluating voltage linearity, we hope that the smaller the value of α, the better, indicating the dependence of capacitance density on voltage. It can be seen from Fig. 5d and Fig. 5e that the overall variation of the dielectric layer with voltage at different frequencies is consistent, and the overall variation range is small.

At present, the prevailing method to explain the variation of capacitance density with bias voltage is usually related to the space charge effect [Journal of Applied Physics. 2001, 290, 1501-1508; Applied Physics Letters. 2007, 91, 172903]. For instance, enhanced electron injection has been suggested for the positive/negative capacitance density-voltage effect in Al2O3 [IEEE Electron Device Letters. 2011, 32, 384-386] and HfO2 [2020 IEEE 70th Electronic Components and Technology Conference (ECTC). 2020, 2139-2144]. Another reason for the different variation trends may be related to the thickness of the amorphous structures prepared by low-temperature ALD deposition. As we all know, capacitance density is significantly sensitive to the dielectric (Al2O3) thickness due to the relatively low dielectric constant of Al2O3. As can be seen from Fig. 5a-c, the 12.5 nm and 25 nm thick Al2O3 MIM capacitors show a slight change with voltage, while the 50 nm thick Al2O3 MIM capacitors remain almost unchanged at the same voltage bias. The sentences “The reason for the different variation trends may be related to the thickness of the amorphous structures prepared by low-temperature ALD deposition. As is well-know, capacitance density is significantly sensitive to the dielectric (Al2O3) thickness due to the relatively low dielectric constant of Al2O3. It can be seen from Fig. 5d-f, the 12.5 nm and 25 nm thick Al2O3 MIM capacitors show a slight change with voltage, while the 50 nm thick Al2O3 MIM capacitors remain almost unchanged at the same voltage bias.” was added on page 11.

Reviewer #3: In this paper, the authors present the fabrication and the dielectric properties of metal-insulator-metal (MIM) capacitors with Al2O3, TiO2, or HfO2 dielectrics obtained by atomic-layer-deposition. They characterize the dielectric films using ellipsometry, XPS, and AFM and systematically study the influence of dielectric material and layer thickness on the performance of MIM capacitors. They find that the Al2O3-based dielectric capacitor exhibits superior electrical performance over HfO2 and TiO2

Although of limited novelty, the study is interesting and can serve for the potential application in integrated passive devices. The manuscript is rather clear and well organized. The results and the related discussion are convincing.

The paper can be suitable for publication after a revision. Here are some points that requires attention:

Response: We acknowledge the reviewer for his/her professional evaluation of our work. We also thank him/her for bringing up valuable comments, which help us improve the results. In the following, we reply one by one to the referee’s questions.

  1. In the introduction: “In recent years, a great number of researches have been devoted to developing energy and capacitance densities in energy storage fields, especially for supercapacitors (SCs) [9, 10].” I suggest adding this recent paper on supercapacitors: https://doi.org/10.1002/admi.202100149.

Response: We thank the reviewer’s suggestions and comments, which make our paper better. In the revised version of the manuscript, we added the recommended article as ref.[11]. The sentence “In recent years, a great number of researches have been devoted to developing energy and capacitance densities in energy storage fields, especially for supercapacitors (SCs) [9-11].” was added on page 2.

  1. “Besides, the prepared low-temperature ALD dielectric layers are amorphous, which can reduce the leakage of leakage current along the edge of the grain boundaries [26]” I guess there is a typo in this sentence.

Response: We are grateful for the reviewer’s suggestion. We modified the sentence, “Besides, the prepared low-temperature ALD dielectric layers are amorphous, which can reduce the leakage current along the edge of the grain boundaries [27]. ” on page 10.

  1. “As it was noticed that there are certain white defects in the film itself after ALD deposition, which are mainly due to the compatibility of different dielectric layers with process parameters during deposition.” Can the authors better define these defects and explain their origin? Besides these white defects seem not to be present in the HfO2 dielectric material.

Response: We appreciate the reviewer for the detailed comment. In our original work, we defined the white spot in the AFM image as a “defect”, which might cause misleading, in fact, the white spot in the AFM just reflects a peak in the film topography roughness there. More white spots also reflect the uniformity of the prepared film to a certain extent. Furthermore, the related discussion “As it was noticed that there are certain white “defects” in the film itself after ALD deposition, and these white “defects” actually refer to the peaks of the surface undulation of the dielectric layer. The defects of the TiO2 film in Fig. 3b are more pronounced, indicating that the film quality of TiO2 is inferior to that of Al2O3 and HfO2, which is mainly due to the lower deposition temperature.” was added on page 7 in our revised main text.

We sincerely thank the reviewer for this valuable comment that will motivate us to think deeply and formulate rigorously in our future work.

  1. “The J-V curve can be divided into two regions at 4.6 V, where the leakage current densities increase sharply with the bias voltage (breakdown voltage). This can be explained by different conductive mechanisms, namely Poole-Frenkel emission and Schottky emission at low and high electric fields [5, 35, 38].” Poole-Frenkel and Schottky emission are only mentioned here but highlighted in the abstract. The authors should elaborate more on this point and possible show fits of the JV dependencies according to the recalled mechanisms.

Response: We really appreciate the professional opinion of the reviewer. Indeed, the Poole-Frenkel mechanism and Schottky mechanism involved in the J-V curve are actually generally accepted explanation mechanisms, but we did not explain them in detail, we made further explanation in the text according to the reviewer’s suggestion. However, we do not think it is necessary to further verify the correctness of the mechanism by curve fitting through the Poole-Frenkel mechanism, because similar curves fitting have been done in the literatures [Micromachines. 2018, 9, 69; Solid State Electronics. 2013, 79, 218-222] we cited, and thus the conclusion in our work is convincing. The sentences “The J-V curve can be divided into two regions at 4.6 V, where the leakage current densities increase sharply with the bias voltage (breakdown voltage). This can be explained by different conductive mechanism of leakage current, namely Schottky emission and Poole-Frenkel emission at low and high electric fields [5, 36, 39], which are shown by (1) and (2), respectively.

                                           J=AT2 exp[- (qS - βSE1/2)/kT]                                       (1)

                                          J=CE exp[- (qPF - βPFE1/2)/kT]                                     (2)

Where A and C are constants, T is the temperature in kelvin, 298 K, E is the electric field, E=U/d, q is the electron charge, S is the barrier height of the interface between the dielectric and injecting electrode for Schottky emission, and PF is the trap height in the dielectric for PF emission, k is the Boltzmann constant, βS and βPF are (q3/πε0n2)1/2 and (q3/πε0n2)1/2, respectively, in which ε0 is a permittivity in vacuum, n is the refractive index.

Equations (1) and (2) are fitted to the experimental data and extract the n value of the refractive index from the slope of the fitted curve, which further indicate that the conduction mechanism of the MIM capacitor is dominated by PF emission at high electric region. In fact, the neutral electron traps in high-k materials are abundantly generated under an applied electric field, resulting in an increase in leakage current. Hence, the leakage current is dominated by the trap-assisted-tunneling at low electric field region. ” was added on page 9.

We sincerely thank the reviewer for this valuable comment that will motivate us for in-depth thinking and investigation in our work.

  1. “As can be seen, the capacitance density of Al2O3 MIM capacitors with different thicknesses (12.10 nm, 24.72 nm and 49.73 nm) exhibit a slight degradation from 0 to 1 MHz, and the values only range from 0.07 to 1.38 fF·μm-2, which reflects the excellent dielectric characteristic with frequencies.” The quoted values 0.07 to 1.38 fF·μm-2 do not seem to match what is reported in plots 5a-c or it is not clear what they are.

Response: We are grateful for the reviewer’s suggestion. Indeed, what we originally wanted to express was that Al2O3 MIM capacitors with different thicknesses exhibited a change in capacitance density from 0 to 1 MHz, and the corresponding change value for 12.10 nm Al2O3 is the difference between the value (14.40146 fF·μm-2) at 0 Hz  and the capacitance value (13.01773 fF·μm-2) at 1 MHz, and the result is 1.38 fF·μm-2, With the same calculation method, for 24.72 nm Al2O3, the difference between the value (8.32896 fF·μm-2) at 0 Hz and the capacitance value (7.8101 fF·μm-2) at 1 MHz is 0.51886 fF·μm-2, and for 49.73 nm Al2O3, the difference between the value (4.5897 fF·μm-2) at 0 Hz and the capacitance value (4.51764 fF·μm-2) at 1 MHz is 0.07 fF·μm-2, so the change range is from 0.07-1.38 fF·μm-2. But in fact, this expression is not professional and clear, and as suggested by the reviewer, we give the maximum change value of each capacitance-frequency curve one by one, so that the given values more closely match with Fig. 5d and Fig. 5e. The sentences “As can be seen, the capacitance density of Al2O3 MIM capacitors with different thicknesses (12.10 nm, 24.72 nm and 49.73 nm) exhibit a slight degradation from 0 to 1 MHz, and the maximum change values are only 1.45, 0.54 and 0.17 fF·µm-2, which reflects the excellent dielectric characteristic of Al2O3 MIM capacitors with frequencies. Nevertheless, under the same preparation conditions, the capacitance densities of TiO2 vary in the range of 5.70, 3.48 and 3.20 fF·µm-2, respectively. For HfO2, the corresponding values are 2.51 and 2.03 fF·µm-2. Compared with Al2O3 dielectric films, neither TiO2 nor HfO2 are relatively satisfactory results. ” was added on page 10.

  1. “This can be concluded that the large band gap energy of barrier type Al2O3 (8.9 eV), while the band gap energies of TiO2 and HfO2 are 3.2 eV and 5.6 eV [22].” I am not sure that I understand the meaning of this sentence which seems to me incomplete. Please rephrase it.

Response: We thank the reviewer’s suggestions and comments, which make our paper better. Indeed, this sentence is incomplete and the reader cannot understand the meaning of this sentence, so we revised this sentence. The sentence “This can be concluded that the large band gap energy (8.9 eV) of the larger barrier type Al2O3 brings excellent frequency stability, while the band gap energies of TiO2 and HfO2 are only 3.2 eV and 5.6 eV, respectively [23].” was added on page 10.

  1. “It can be inferred from Fig. 5d and Fig. 5e that the overall capacitance density of the 12.5 nm thickness Al2O3 dielectric capacitor decreases slightly with the bias voltages, and the 25 nm thickness capacitor increases slightly, whereas the increase and decrease of the maximum values are only 1.19 fF·μm-2 and 0.25 fF·μm-2.” Have the authors any explanation of this different trend of the capacitance vs bias?

Response: We really appreciate the helpful comments. In fact, for dielectric capacitors, the relationship of capacitance with bias voltage is

C(V)=C0(αV2+βV+1)

(C(V)-C0)/C0=[αV2+βV]ppm

Where C0 represents the zero-bias capacitance, α and β are the quadratic and linear voltage coefficients, respectively. Among them, quadratic voltage coefficient α is the most critical parameter for evaluating voltage linearity, we hope that the smaller the value of α, the better, indicating the dependence of capacitance density on voltage. It can be seen from Fig. 5d and Fig. 5e that the overall variation of the dielectric layer with voltage at different frequencies is consistent, and the overall variation range is small.

At present, the prevailing method to explain the variation of capacitance density with bias voltage is usually related to the space charge effect [Journal of Applied Physics. 2001, 290, 1501-1508; Applied Physics Letters. 2007, 91, 172903]. For instance, enhanced electron injection has been suggested for the positive/negative capacitance density-voltage effect in Al2O3 [IEEE Electron Device Letters. 2011, 32, 384-386] and HfO2 [2020 IEEE 70th Electronic Components and Technology Conference (ECTC). 2020, 2139-2144]. Another reason for the different variation trends may be related to the thickness of the amorphous structures prepared by low-temperature ALD deposition. As we all know, capacitance density is significantly sensitive to the dielectric (Al2O3) thickness due to the relatively low dielectric constant of Al2O3. As can be seen from Fig. 5a-c, the 12.5 nm and 25 nm thick Al2O3 MIM capacitors show a slight change with voltage, while the 50 nm thick Al2O3 MIM capacitors remain almost unchanged at the same voltage bias. The sentences “The reason for the different variation trends may be related to the thickness of the amorphous structures prepared by low-temperature ALD deposition. As is well-know, capacitance density is significantly sensitive to the dielectric (Al2O3) thickness due to the relatively low dielectric constant of Al2O3. It can be seen from Fig. 5d-f, the 12.5 nm and 25 nm thick Al2O3 MIM capacitors show a slight change with voltage, while the 50 nm thick Al2O3 MIM capacitors remain almost unchanged at the same voltage bias.” was added on page 11.

Round 2

Reviewer 3 Report

All the concerns/comments are considered carefully, and the authors have modified the manuscript accordingly. The modifications indeed improved the manuscript quality and readability. The findings are appropriately represented and easily understandable. I recommend the current version of the manuscript for publication.